# An Improved Data Processing Algorithm for Spectrally Resolved Interferometry Using a Femtosecond Laser

**DOI:** 10.3390/s24092869

**Published:** 2024-04-30

**Authors:** Tao Liu, Hiraku Matsukuma, Amane Suzuki, Ryo Sato, Wei Gao

**Affiliations:** Precision Nanometrology Laboratory, Department of Finemechanics, Tohoku University, Sendai 980-8579, Japan; liu.tao.q8@dc.tohoku.ac.jp (T.L.); amane.suzuki.t5@dc.tohoku.ac.jp (A.S.); ryo.sato.b8@tohoku.ac.jp (R.S.); i.ko.c2@tohoku.ac.jp (W.G.)

**Keywords:** absolute distance measurement, spectrally resolved interferometry, inverse Fourier transform

## Abstract

Spectrally resolved interferometry utilizing a femtosecond laser is widely employed for absolute distance measurement. However, deviations in the output time pulse of the conventional algorithm through inverse Fourier transform are inevitable. Herein, an improved data processing algorithm employing a time-shifting parameter is proposed to improve the accuracy of spectrally resolved interferometry. The principle of the proposed time-shifting algorithm is analyzed theoretically after clarifying the deviation source of the conventional algorithm. Simulation and experimental work were conducted to indicate the improvement in the accuracy of the output absolute distance. The results demonstrated that the proposed algorithm could reduce the deviation of output distances towards the reference values, reaching 0.58 μm by half compared to the conventional algorithm. Furthermore, the measurement uncertainty was evaluated using the Guide to the Expression of Uncertainty in Measurement (GUM), resulting in an expanded uncertainty of 0.71 μm with a 95% confidence.

## 1. Introduction

With the development of science and technology, absolute distance measurement is widely utilized in semiconductor manufacturing, electromechanical systems, and satellite formation flying [1,2,3,4,5]. Benefiting from the properties of non-contact and high-resolution, absolute distance measurements based on optical methods were generally researched [6,7,8]. Time-of-flight is a typical optical method that determines the target distance by measuring the time interval between the emission of an optical pulse and the reception of the reflective pulse [9]. However, the resolution of this approach is limited to several millimeters, due to constraints such as the response time of photodetectors and the time width of the laser pulse. On the other hand, multi-wavelength interferometry for absolute distance measurement was developed by Wyant et al. [10,11,12]. This proposal significantly extends the unambiguous range compared to the conventional Michelson interferometer using only one wavelength, which is operated by utilizing a virtual synthetic wavelength generated from the beat signal of two wavelengths [13]. Nevertheless, multiple different wavelength laser sources and their alignment make this absolute distance measurement system complicated and expensive. Since the optical frequency comb (OFC) was incepted at the end of the 20th century, absolute distance measurement with high-resolution was further developed [14,15,16]. An optical frequency comb is composed of numerous narrow linewidth wavelengths over a broad optical spectral range in the frequency domain, and a femtosecond pulse sequence in the time domain [17,18]. Each frequency component of the OFC can be precisely indicated once the repetition frequency and carrier-envelope frequency are locked to the radio frequency standards [19,20]. The gap between optical frequency and radio frequency is bridged by the OFC, resulting in the promotion of optical frequency measurement [19,21], ultra-precision spectroscopy [22,23], absolute distance [24,25], angle measurement [26,27,28], etc. Moreover, numerous absolute distance measurement techniques have been developed based on the OFC, including synthetic wavelength interferometry (SWI) [29,30,31,32], multiwavelength interferometry (MWI) [33,34,35], dual-comb interferometry [36,37,38], and spectrally resolved interferometry (SRI) [39,40,41,42]. Benefiting from a compact optical configuration with high-resolution, spectrally resolved interferometry, which operates based on interferograms over a broad spectrum, has been widely researched over the past two decades.

However, some shortcomings still afflict SRI, such as dead zone and intermediate accuracy. The authors previously proposed a spectral fringe algorithm to reduce the dead zone to its theoretical minimum working distance by removing the influence of the source spectrum and subsequently proposed a high-order-angle algorithm to further breakthrough the dead zone over the theoretical limitation to near zero [43,44]. On the other hand, many researchers also made great contributions to increasing the accuracy of the time pulse position of SRI. The time pulse is obtained by a direct inverse Fourier transform of the interference signal, and its peak position should be equal to the time delay caused by the target distance. In the first stage, the obtained time pulse is a broad one due to the limited time resolution, which is determined by the frequency width. Many researchers tried to increase the accuracy of the time pulse by increasing the employed spectral width or locating the precise peak position by curve fitting. For instance, Jang et al. developed a spectrally resolved interferometer powered by a soliton microcomb, and a polynomial fitting method was employed to extract the time delay of the inverse Fourier transform more accurately to output a target distance with higher accuracy [45,46]. Similarly, Wang et al. proposed a chip-scaled soliton microcomb, a three-point fitting method was required in locating the time-domain peak precisely [47]. However, these authors ignored the necessity of truncating the interference signal into integer periods before performing the inverse Fourier transform, which is an indispensable operation for generating the time pulse accurately, making the time pulse not an ideal sharp shape and limited by the employed frequency width. In the second stage, a truncation to make the interference signal into integer periods is conducted before the inverse Fourier transform as the authors previously proposed and named the truncated-spectrum algorithm [43]. A sharper time pulse can be obtained, and its position is not limited by the frequency width, because the time pulse must be located at integer times of the time resolution for a truncated period signal based on the principle of Fourier transform [48,49]. However, the accuracy of the truncated-spectrum algorithm suffers from the limited spectral resolution, resulting in the truncation points, i.e., the minimum and maximum frequencies of the truncated interference signal are not the ideal ones. The spectral resolution is determined by the used spectrometer and cannot be improved to zero, refrained from the modern instrumentation technology. Hence, an improved algorithm is required to further enhance the accuracy of spectrally resolved interferometry.

In this paper, an improved data processing algorithm for spectrally resolved interferometry is proposed after clarifying the deviation source of the time delay pulse, which can achieve the time delay pulse with a higher accuracy. This proposed algorithm is named the time-shifting algorithm because a time shift parameter *t_s_* is utilized in it. The effectiveness of this algorithm is verified by the simulation and experimental results. The absolute distance measurement deviation of the proposed algorithm is compared with the conventional algorithm. In addition, the uncertainty of the distance measurements is estimated based on the “Guide to the expression of Uncertainty in Measurement” (GUM) [50].

## 2. Principles

Figure 1 shows the optical setup of a Michelson-type spectrally resolved interferometer irradiated by a femtosecond laser. The spectral interference signal can be detected by a spectrometer and its intensity can be simply described as follows:(1)$$I(f_{k})=S(f_{k})\cdot \left[1+A\cdot \cos\left(2\pi f_{k}\tau \right)\right]$$
where *S*(*f_k_*) is the spectral intensity of the femtosecond laser source and *A* is a parameter representing the spectral intensity difference of the reference and measurement beam. *τ* is the time delay caused by the optical path difference *L* between the reference and measurement paths, and it can be calculated by:(2)$$\tau =\frac{2nL}{c}$$
in which *n* is the refractive index of air, and *c* represents the speed of light.

In the conventional data processing algorithm of the spectrally resolved interferometry, the spectral interference signal is directly inverse Fourier transformed into a time-domain function *i*(*t_i_*) as follows:(3)$$i\left(t_{i}\right)=\frac{1}{N}\sum_{k=0}^{N-1}I\left(f_{k}\right)\cdot e^{\frac{j2\pi ki}{N}}=s\left(t_{i}\right)⊗\left[\delta \left(t_{i}\right)+\frac{A}{2}\delta \left(t_{i}-\tau \right)+\frac{A}{2}\delta \left(t_{i}+\tau \right)\right]=s\left(t_{i}\right)+\frac{A}{2}\cdot s\left(t_{i}-\tau \right)+\frac{A}{2}\cdot s\left(t_{i}+\tau \right)$$
where *N* is the sampling number, *δ*(*t_i_*) represents a unit impulse function, and *s*(*t_i_*) is the inverse Fourier transform of the source spectrum *S*(*f_k_*). Three time pulses can be observed in the results of the inverse Fourier transform, with their peaks located at −*τ*, 0, and *τ*, respectively. The coincidence of two pulses located at 0 and *τ* with extremely small *τ* can remarkably worsen the minimum working distance of the spectrally resolved interferometry, which can be solved by removing the central time pulse and sharping the width of the time pulses. The authors previously proposed a spectral fringe algorithm to achieve it, specifically, removing the upper and lower envelopes of the detected spectral interference signal in Equation (1), followed by normalization [44]. A modified spectral interference signal *I_m_*(*f_k_*) can subsequently be generated and described as:(4)$$I_{m}\left(f_{k}\right)=\cos\left(2\pi f_{k}\tau \right)=\left(\frac{I\left(f_{k}\right)}{S\left(f_{k}\right)}-1\right)\cdot \frac{1}{A}$$

Hence, the inverse Fourier transform of this modified interference signal can be expressed as the modified time function *i_m_*(*t_i_*):(5)$$i_{m}(t_{i})=\frac{1}{N}\sum_{k=0}^{N-1}I_{m}\left(f_{k}\right)\cdot e^{\frac{j2\pi ki}{N}}=\frac{1}{2}\cdot \delta \left(t_{i}+\tau \right)+\frac{1}{2}\cdot \delta \left(t_{i}-\tau \right)$$

A time pulse *τ*_1_ is then extracted using a time window centered at *τ*, and the target distance can be directly calculated using the formula *L* = *τ*_1_ · *n* · *c*/2, where *τ*_1_ is the position value of this selected time pulse in the horizontal axis. The parameter *τ*_1_ is referred to as the measured time delay, which is determined by the inverse Fourier transform results. Furthermore, the parameter *τ* is regarded as the real-time delay caused by a target distance. To distinguish the measured time delay and the real-time delay, two different parameters of *τ* and *τ*_1_ are utilized in this paper. It is evident that the accuracy of the final calculated distance strongly relies on that of the selected time pulse *τ*_1_. Ideally, the selected time pulse *τ*_1_ should be equal to the real-time delay *τ*. However, there is always some deviation between the selected time pulse *τ*_1_ and the real-time delay *τ* owing to the necessary operation of the inverse Fourier transform in the conventional algorithm. Therefore, eliminating the deviation between *τ*_1_ and *τ* is a crucial strategy for achieving high precision in the measurement of the target distance using spectrally resolved interferometry.

Based on the theory of discrete Fourier transform, the equivalence of discrete and continuous Fourier transform requires that the periodic function must be truncated over exactly one period (or integer multiple periods) [48]. The authors previously implemented the truncation process based on the local maximum of the interference signal. This approach was chosen because the phase value is consistently either 0 or 2π for a cosine function at local maxima. As a result, integer periods of the interference signal can be truncated, making it suitable for a subsequent inverse Fourier transform. When the interference signal reaches its local maximum, constructive interference is conducted, resulting in the minimum *f*_1_ and maximum truncated frequency *f*_2_ satisfying the following equation:(6)$$L=\frac{1}{2}\cdot p\cdot \frac{c}{f_{1}}=\frac{1}{2}\cdot \left(p+q\right)\cdot \frac{c}{f_{2}}$$
in which *p* represents a random positive integer determined by the target distance *L*, and *q* represents the period number of the truncated spectrum. The integer *p* can be calculated as follows:(7)$$p=q\cdot \frac{\lambda_{2}}{\lambda_{1}-\lambda_{2}}$$
in which *λ*_1_ and *λ*_2_ represent the corresponding wavelength of truncated frequencies *f*_1_ and *f*_2_, calculated by *λ* = *c*/*f*.

Based on the principle of inverse Fourier transform, the time resolution ∆*t* of an ideally truncated spectrum can be calculated by the reciprocal of frequency width as follows:(8)$$\Delta t=\frac{1}{f_{2}-f_{1}}$$

The resolution of the output distance, i.e., the distance resolution ∆*L*, can be easily deduced by the formula of ∆*L* = ∆*t* · *c*/2 as follows:(9)$$\Delta L=\frac{c}{2}\cdot \Delta t=\frac{c}{2}\cdot \frac{1}{f_{2}-f_{1}}=\frac{1}{2}\cdot \frac{\lambda_{1}\times \lambda_{2}}{\lambda_{1}-\lambda_{2}}$$

It is worth noting that both the time resolution ∆*t* and distance resolution ∆*L* are not fixed by the employed frequency width and vary with the target distance, benefitting from the truncated spectrum. Based on Equations (6)–(9), the target distance *L* can be re-calculated as follows:(10)$$L=\frac{1}{2}\cdot p\cdot \frac{c}{f_{1}}=\frac{1}{2}\cdot q\cdot \frac{\lambda_{1}\times \lambda_{2}}{\lambda_{1}-\lambda_{2}}=q\cdot \Delta L=\frac{c}{2}\cdot q\cdot \Delta t$$

It can be easily found that the target distance *L* must be $\frac{c}{2}\cdot q$ times of the time resolution ∆*t*. As a result, any error in the time resolution can be amplified by $\frac{c}{2}\cdot q$ times in the final output distance. For instance, a deviation of 0.15 μm can be found in the final output distance even if there is only a deviation of 1 ps in the time resolution, assuming *q* = 1. Therefore, it is necessary to obtain the time resolution with high accuracy by truncating the spectrum precisely.

However, due to the limited spectral resolution of spectrometers, it is impossible to truncate the interference signal into integer periods without any error, as illustrated in Figure 2b. A comparison between the ideally truncated and non-ideally truncated spectrum is shown in Figure 2, in which *f*_1_ and *f*_2_ in Figure 2a represent the minimum and maximum frequency of the ideally truncated spectrum. Meanwhile, *f*_1_′ and *f*_2_′ in Figure 2b represent the minimum and maximum frequency of the non-ideally truncated spectrum. It is worth noting that any inaccuracies in the minimum and maximum frequency of the truncated interference signal can introduce errors in the time resolution ∆*t* of the inverse Fourier transform results. This ultimately yields a deterioration in the accuracy of the output distance results.

The time resolution ∆*t*′ of a non-ideally truncated spectrum can be calculated as follows:(11)$$\Delta t^{\prime }=\frac{1}{f_{2}^{\prime }-f_{1}^{\prime }}$$

Deviations between the ideally and non-ideally truncated frequencies result in errors occurring in the time resolution in the inverse Fourier transform. In other words, the value of ∆*t*′ is not equal ∆*t*, which is the time resolution of an ideally truncated spectrum. Meanwhile, the selected time pulse *τ*_1_ is located at integer periods of the time resolution, i.e., *τ*_1_ = *q* × ∆*t*′, in which *q* is an arbitrary integer, and its value is equal to the integer period number of the cosine item in the modified interference signal. Generally, more than one integer period is employed in the spectrally resolved interferometry, making the deviation of the selected time pulse *τ*_1_ towards the real-time delay *τ* be magnified several times for the final output distance. Eliminating the deviation between the selected time pulse *τ*_1_ and the real-time delay *τ* or just directly positioning the value of real-time delay *τ* is a crucial approach to precisely achieve the target distance using spectrally resolved interferometry.

Although there is a deviation between the selected time pulse *τ*_1_ and the real-time delay *τ*, this deviation is further smaller than the time resolution. Specifically, the real-time delay *τ* is always located in the vicinity of the selected time pulse *τ*_1_. Therefore, it is feasible to determine the real-time delay *τ* by comparing the inverse Fourier transform results of all available time points near the selected time pulse *τ*_1_ within the interval [*τ*_1_ − ∆*t*′, *τ*_1_ + ∆*t*′]. Based on the theory of Fourier transform, the absolute value of the inverse Fourier transform results should reach its maximum of 0.5 at the real-time delay point *τ*. Meanwhile, the absolute value of the inverse Fourier transform for other time points, especially the selected time pulse *τ*_1_, should be smaller than that of the real-time delay *τ*. The inverse Fourier transform of other time points near *τ*_1_ can be generated by utilizing a time-shifting parameter *t_s_*, which can be calculated by:(12)$$i_{m}(t_{i}+t_{s})=\frac{1}{N}\sum_{k=0}^{N-1}I_{m}\left(f_{k}\right)\cdot e^{j2\pi \cdot \left(\frac{i}{f_{2}^{\prime }-f_{1}^{\prime }}+t_{s}\right)\cdot \left[\frac{k\cdot \left(f_{2}^{\prime }-f_{1}^{\prime }\right)}{N}\right]}=\frac{1}{2}\cdot \delta \left(t+\tau_{1}+t_{s}\right)+\frac{1}{2}\cdot \delta \left(t-\tau_{1}+t_{s}\right)$$

Time points near the selected time pulse *τ*_1_ can be obtained by dividing the interval of [*τ*_1_ − ∆*t*′, *τ*_1_ + ∆*t*′] into 2*M* pieces, in which *M* is the segmentation number as shown in Figure 3. The time-shifting parameter *t_s_* is a sequence of time points within the interval and can be expressed as follows:(13)$$t_{s}=\tau_{1}+s\cdot \frac{\Delta t^{\prime }}{M}\;s=0,\pm 1,\ldots ,\pm M$$
where *s* is an arbitrary integer and ∆*t*′ is the time resolution in the inverse Fourier transform results. The segmentation number *M* needs to be large enough to ensure that the real-time delay can be found during this operation.

Finally, the optimum value of the time-shifting parameter *t_s_*_0_ can be determined by searching for the maximum absolute value of the inverse Fourier transform among all available time points. The optimum time-shifting parameter *t_s_*_0_ satisfies *τ* = *τ*_1_ + *t_s_*_0_, and an improved output distance can be calculated by:(14)$$L=\frac{c}{2n}\cdot \left(\tau_{1}+t_{s0}\right)$$

## 3. Simulation and Experiment Results

### 3.1. Simulation Results

A modified interference spectrum *I_m_*(*f_k_*) defined in Equation (4) with a target distance of 1 mm, and a spectral frequency ranging from 191.7 THz to 195.2 THz with a sampling frequency of 10 GHz was exploited in a simulation to illustrate the advantages of the proposed time-shifting algorithm on increasing the accuracy of spectrally resolved interferometry. The segmentation number *M* for the time-shifting algorithm was set as 400. Figure 4 illustrates the data processing procedure of the proposed time-shifting algorithm and a comparison of the inverse Fourier transform results between the time-shifting algorithm and the conventional one.

For the conventional algorithm, a time pulse *τ*_1_ can be selected from the inverse Fourier transform results using a time window and the target distance can be determined by *L* = 0.5 · *c* · *τ*_1_, as shown in Figure 4c. The accuracy of the selected time pulse *τ*_1_ is influenced by the truncating operation and can be improved by implementing the proposed time-shifting algorithm. Compared with the conventional algorithm, the proposed time-shifting algorithm is capable of achieving the inverse Fourier transform results of more time points near the selected time delay *τ*_1_ with the help of time-shifting sequence *t_s_*, as illustrated in Figure 4e. Moreover, the real-time delay *τ* can be found by searching for the optimum time-shifting parameter *t_s_*_0_, which can reach the maximum absolute value of the inverse Fourier transform results, as shown in Figure 4f. The real-time delay *τ* caused by a target distance of 1 mm can be ascertained as 6.671 ps, while the selected time delay *τ* based on the conventional algorithm is 6.666 ps with an error of 5 fs towards the real-time delay. However, employing the time-shifting algorithm can significantly diminish this error, resulting in an improved time delay of 6.672 ps by determining the optimum time-shifting parameter *t_s0_*. Finally, an improved distance result can be calculated using the formula *L* = 0.5 · *c* · (*τ*_1_ + *t_s_*_0_). It is worth mentioning that there remains a minor deviation of 1 fs between the improved time delay and the real-time delay, primarily attributed to the calculation errors during the inverse Fourier transform. Nonetheless, the proposed time-shifting algorithm has been verified to be capable of effectively enhancing the accuracy of the final output distance result.

To further illustrate the benefits of the proposed time-shifting algorithm, a simulation was conducted covering a distance range from 500 μm to 1500 μm, with a step size of 1 μm. The spectral frequency is set from 191.7 THz to 195.2 THz, with a sampling frequency of 10 GHz. Figure 5 presents a comparison of simulation results obtained using the proposed time-shifting algorithm and the conventional algorithm.

As shown in Figure 5, compared to the conventional algorithm, the simulated deviation from the reference distance shows an obvious decrease in the time-shifting one. Specifically, the absolute average deviation of the conventional algorithm is 1.01 μm, whereas this value from the time-shifting algorithm is reduced to 0.44 μm, more than two times smaller than that of the conventional one. Meanwhile, the standard deviation of the conventional algorithm decreases from 1.29 μm to 0.45 μm when using the time-shifting algorithm. Therefore, the effectiveness of the proposed time-shifting algorithm can be verified by these simulation results. It is worth noting that the deviation of the time-shifting algorithm can be further decreased to zero theoretically by increasing the segmentation number *M* of the time interval. However, an increasing segmentation number yields an increasing computation time to operate the discrete inverse Fourier transform. This trade-off relationship cannot be ignored, and a suitable value of the segmentation number should be chosen when utilizing the proposed time-shifting algorithm. The computing time for one target distance using the proposed time-shifting algorithm is approximately 80 s, in which 400 times of an inverse Fourier transformation with different time shifts are conducted.

### 3.2. Experimental Setup

Figure 6 shows a schematic of the experimental setup for feasibility measurements of the proposed time-shifting algorithm based on spectrally resolved interferometry. A mode-locked femtosecond fiber laser source with a center frequency *f_c_* of 192.175 THz and a repetition frequency *f*_rep_ of 100 MHz, is projected into the beam splitter via a single-mode fiber. The reflected and transmitted light of the beam splitter is then reflected by two square-protected silver mirrors, employed as reference and measurement beams, respectively. The reference mirror is kept stable, while the measurement mirror is translated linearly along the measurement beam through a single-axis motorized stage (Suruga Seiki, KXC04015-CA, Shizuoka, Japan). The minimum moving step of this stage is 0.1 μm. Moreover, the position of the measurement mirror is monitored by a commercial laser encoder (Renishaw, RLD10-3R, Wotton-under-Edge, UK), whose target retroreflector is mounted on the motorized stage. The optical interference signal is analyzed by an optical spectrum analyzer (Yokogawa, AQ6370D, Tokyo, Japan) connected by an optical fiber.

### 3.3. Experimental Results and Discussion

In the experiment, the optical interference signal within a range of 191.7 THz to 195.2 THz was recorded by an optical spectrum analyzer (OSA) with a resolution of 0.02 nm. The laser encoder was set to zero at the real zero position to calibrate the moving distance of the measurement mirror, whose output displacement is employed as the reference displacement. However, this device can only determine the relative displacement from position A to position B. In other words, the employed commercial laser encoder cannot directly output the absolute displacement between the measured mirror and the real zero position. If the output of the laser encoder is desired to be worked as the reference displacement with an absolute type, the real zero position must be determined in advance and set as the zero for the laser encoder. A precise determination of the real zero position is the baseline to generate a reliable absolute type of reference distance. The real zero position is defined as the length of the measurement and reference arms are equal to each other in this position. In our experiment, the real zero position is determined by searching for which step the motorized stage is at when the power summation of all available wavelengths in the spectrum can reach the maximum. It operates based on the assumption that the phase difference caused by the optical path difference is zero at this real zero position and constructive interference can be generated for all the wavelengths because the length of the measurement and reference arms are equal at this position. However, the accuracy of the real zero position is limited by the minimum available moving step of the motorized stage as ±0.2 μm (the displacement caused by the target mirror is twice the movement of the target mirror). In other words, there is a deviation between the ideal real zero position and the determined zero position. Due to the target mirror being moved by the motorized stage with a minimum displacement of 0.1 μm and cannot be further reduced to the infinite minimum, the real zero position is nearly impossible to reach without any error. Therefore, a small deviation existed in the reference displacement resulting in a small systematic measurement error. However, this error does not affect the performance evaluation of the proposed time-shifting algorithm and reflects the advantage of the spectrally resolved interferometer on the absolute distance measurement in contrast to the commercial laser encoder.

The detected interference spectrum was processed both by the conventional and the proposed time-shifting algorithm, and the data processing procedure is explained in Figure 7. The raw interference spectrum with a reference distance of 500.54 μm as well as the source spectrum are shown in Figure 7a. Preprocessing of the raw interference spectrum is required to remove the noise and other disturbances. The preprocessing procedures are constructed by the following three steps. First, remove the influence of the source spectrum *S*(*f_k_*) by dividing the source spectrum directly. Then, performing a normalization to eliminate the effect of the direct current term ‘1’ and parameter *A* in the divided spectrum, which is obtained from the last step. Finally, a spline interpolation is conducted to the normalized spectrum, and its spectral resolution can be increased based on it. Although a smaller spectrum resolution can be achieved by using a smaller interpolation frequency, the required computer memory and computing time for the following inverse Fourier transformation are both dramatically increased. This trade-off relationship cannot be ignored. Consequently, we employed a spline interpolation with an interpolation frequency of 1 GHz to interpolate the interference spectrum, resulting in a deviation during the following truncation that cannot be neglected. Figure 7b shows the spectrum after truncating the preprocessed spectrum into integer periods.

Two pulses in the time domain were achieved after operating an inverse Fourier transform on the truncated signal, in which the time delay can be selected by a time window as *τ*_1_ = 3.344 ps. For the conventional algorithm, the measurement distance can be directly calculated by *L* = 0.5 · *c* · *τ*_1_ = 501.25 μm with a deviation of 0.74 μm towards the reference distance. However, employing the proposed time-shifting algorithm can improve the result with higher accuracy by comparing the inverse Fourier transform results of all available time points near the selected time pulse, which is operated by adding a time shift parameter *t_s_*, as shown in Figure 7e. The three black points represent the original results obtained by the conventional algorithm, while the red points represent the results of the proposed time-shifting algorithm. Figure 7f illustrates that a more accurate time pulse can be located by searching for the inverse Fourier transform that has the maximum absolute value. The time delay determined by the time-shifting algorithm was *τ* = 3.340 ps, resulting in a measurement distance of 500.65 μm with a reduced deviation of 0.11 μm.

To further verify the effectiveness of the proposed time-shifting algorithm, the measurement mirror was moved continuously from 500 to 600 μm with a step of 10 μm. The optical interference spectrum was recorded repeatedly ten times by the OSA at a certain position. The measurement results of the conventional and the proposed time-shifting algorithm were compared in Figure 8.

The measurement displacements obtained from both the conventional and the time-shifting algorithms showed a strong agreement with the reference displacement, as depicted in Figure 8. The deviation of each measured displacement from the reference value was determined by averaging the results of ten repetitions of the experiment, and the corresponding standard deviation was also calculated. These standard deviations are presented as the error bars shown in Figure 8. The average deviation for the conventional algorithm was 0.91 μm with a standard deviation of 0.46 μm. However, more accurate distance results could be yielded by the proposed time-shifting algorithm. The average deviation and the standard deviation of measurement results from the time-shifting algorithm were reduced to 0.58 μm and 0.20 μm respectively, which was approximately two times smaller than that of the conventional algorithm. Therefore, the proposed time-shifting algorithm was verified to be capable of increasing the accuracy of spectrally resolved interferometry.

## 4. Uncertainty Analysis

The measurement results are also evaluated using measurement uncertainty analysis based on guides to the expression of uncertainty in measurement (GUM) [50]. The uncertainty of the absolute distance measurement is mainly composed of the refractive index and the uncertainty related to the time pulse locating. The standard uncertainty of the absolute distance measurement can be evaluated by the following equation:(15)$$u_{L}=\sqrt{{\left(c_{n}\cdot u_{n}\right)}^{2}+{\left(c_{\tau_{1}}\cdot u_{\tau_{1}}\right)}^{2}+{\left(c_{t_{s}}\cdot u_{t_{s}}\right)}^{2}+2\cdot c_{\tau_{1}}\cdot c_{t_{s}}\cdot cov\left(\tau_{1},t_{s}\right)}$$
(16)$$c_{n}=\frac{-c\cdot \left(\tau_{1}+t_{s}\right)}{2n^{2}}$$
(17)$$c_{\tau_{1}}=c_{t_{s}}=\frac{c}{2n}$$
where *u_n_*, *u_τ_*_1_, and *u_ts_* are standard uncertainties of *n*, *τ*_1_, and *t_s_*. *c_n_*, *c_τ_*_1_, and *c_ts_* are sensitive coefficients of *u_n_*, *u_τ_*_1_, and *u_ts_*. There is a correlation between the parameter *τ*_1_ and *t_s_* because the time-shifting parameter *t_s_* is employed for compensating the deviation caused by the selected time pulse *τ*_1_. The fourth item of *cov*(*τ*_1_, *t_s_*) in Equation (15) represents the covariance of *τ*_1_ and *t_s_*.

According to the modified Edlén equation, the air refractive index is influenced by temperature, air pressure, and relative humidity [51,52,53]. The combined uncertainty of *u_n_* can be calculated by:(18)$$u_{n}=\sqrt{{\left(c_{t}\cdot u_{t}\right)}^{2}+{\left(c_{p}\cdot u_{p}\right)}^{2}+{\left(c_{h}\cdot u_{h}\right)}^{2}}$$
where *c_t_*, *c_p_*, and *c_h_* are sensitivity coefficients of temperature, pressure, and relative humidity. *u_t_*, *u_p_*, and *u_h_* are the uncertainty of experimental parameters. The laboratory environment was controlled strictly, resulting in a fluctuation of 0.1 K, 0.2 hPa, and 1%, separately. The uncertainty of the refractive index can be calculated to be 1.406 × 10^−8^, as shown in Table 1.

The standard uncertainties for the selected time pulse and time-shifting parameter were evaluated based on the measurement results of the repetitive measurements. The standard uncertainty and the combined uncertainty of the absolute distance measurement are then summarized in Table 2. The expanded uncertainty was 0.71 μm (coverage factor *k* = 2, 95% confidence). It can be easily found that the uncertainty of the distance measurement is primarily attributed to the uncertainty of the selected time pulse and the time-shifting parameter. It is worth noting that the standard deviation of the time-shifting parameter is heavily influenced by that of the selected time pulse because the time-shifting parameter is employed for compensating the deviation caused by the selected time pulse. The correlation between these two parameters was considered in the uncertainty analysis, as the covariance item in Equation (15). Therefore, to increase the accuracy of the absolute distance measurement, the time pulse should be determined with high accuracy, which can be performed by using a high-resolution spectrometer to truncate the interference signal accurately.

## 5. Conclusions

In this paper, we propose a data processing algorithm to improve the accuracy of spectrally resolved interferometry using a femtosecond laser. The principle of the proposed time-shifting algorithm was analyzed theoretically, and simulation was operated to clarify the advantage of this algorithm in increasing the output distance accuracy. A Michelson-type interferometer setup was established to perform absolute distance measurements using spectrally resolved interferometry, and the experimental results obtained from the proposed algorithm and the conventional one were compared. It was verified that employing the proposed algorithm could achieve more accurate measurement results with a lower average deviation of 0.58 μm towards reference distance, which was approximately half smaller than that of the conventional one. The expanded uncertainty of absolute distance measurement was evaluated as 0.71 μm with a 95% confidence based on GUM.

## Figures and Tables

**Figure 1 sensors-24-02869-f001:**
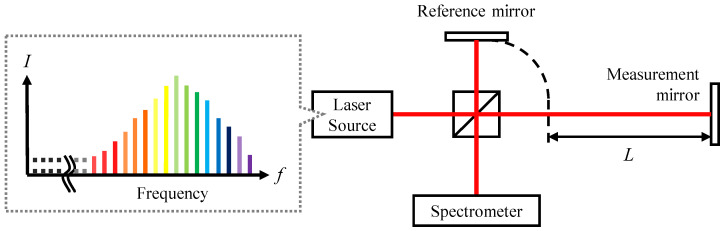
The schematic of a spectrally resolved interferometer using a femtosecond laser source.

**Figure 2 sensors-24-02869-f002:**
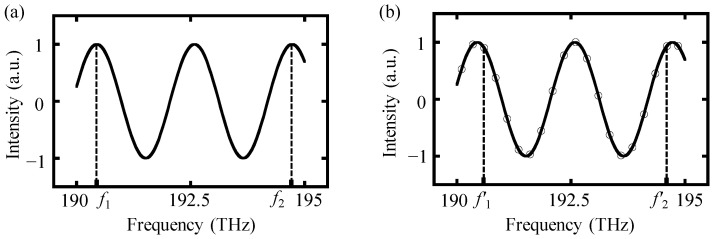
The ideally and non-ideally truncated results of a modified interference spectrum. (**a**) An ideally truncated spectrum and (**b**) a non-ideally truncated spectrum due to the limited spectral resolution.

**Figure 3 sensors-24-02869-f003:**
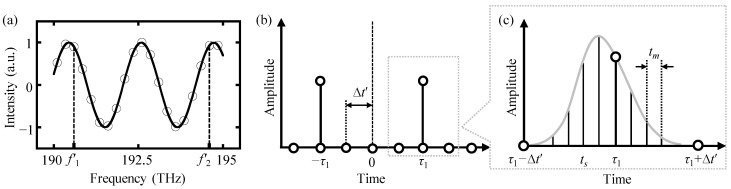
An explanation of the definition of the time-shifting parameter *t_s_* within the time interval of [*τ*_1_ − ∆*t*′, *τ*_1_ + ∆*t*′]. (**a**) A non-ideally truncated interference spectrum; (**b**) inverse Fourier transform results of the spectrum in (**a**); and (**c**) dividing the time interval into 2*M* pieces to generate the time sequence of *t*_s_. The parameter *t_m_* represents the gap between the divided time points and can be calculated by *t_m_* = ∆*t*′/*M*. Hence, the time shift parameter *t_s_* can be determined by *t_s_* = *τ*_1_ + *s* · *t_m_*, in which *s* is an arbitrary integer number no more than *M*.

**Figure 4 sensors-24-02869-f004:**
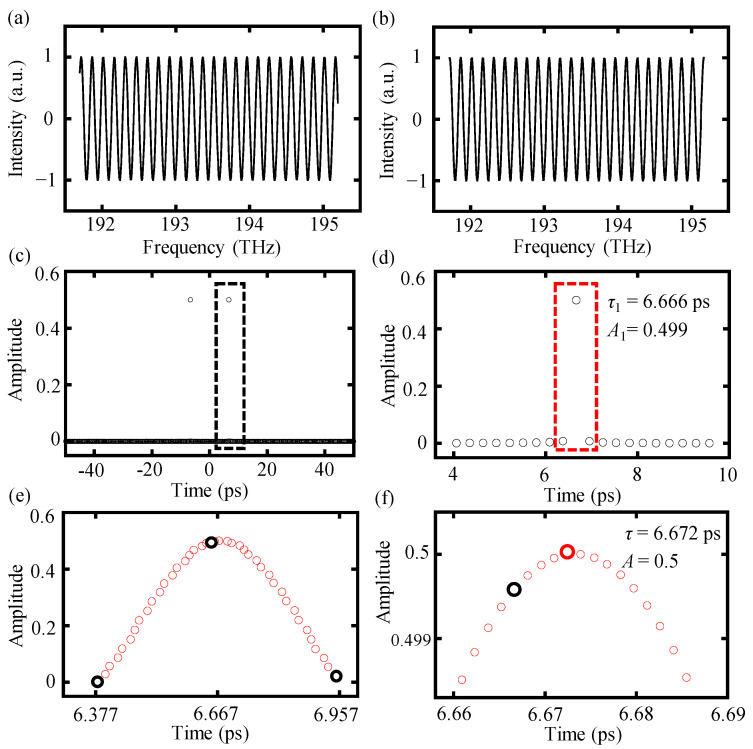
The data processing procedure of the time-shifting algorithm. (**a**) Original modified spectral interference signal *I_m_*(*f_k_*) for a target distance of 1 mm, with the spectral frequency ranging from 191.7 THz to 195.2 THz, (**b**) truncating the interference signal within integer periods, (**c**) inverse Fourier transform results of the truncated signal, and data points in the dot black frame contains the selected time pulse *τ*_1_, and (**d**) magnification of the results in the black frame of (**c**) and the selected time pulse *τ*_1_ value can be checked as 6.666 ps. This time pulse is picked up based on the maximum absolute value of the inverse Fourier transform results among all the available points in the time axis. Furthermore, the red frame highlights the selected time pulse *τ*_1_ and the two adjacent time points, i.e., *τ*_1_ ± ∆*t*′, (**e**) inverse Fourier transform results of the time-shifting algorithm. Three black circles represent the inverse Fourier transform results of the original three time points in the red frame of (**d**), obtained by the conventional algorithm. The middle black circle is the selected time delay pulse *τ*_1_. The red circles represent the inverse Fourier transform results of a sequence of time points near the selected time pulse *τ*_1_, obtained by the time-shifting algorithm, (**f**) amplification of the central region (**e**). The real-time delay *τ*, highlighted as the block red circle, is positioned by searching for the maximum absolute value of the inverse Fourier transform results among the time-shifting sequence.

**Figure 5 sensors-24-02869-f005:**
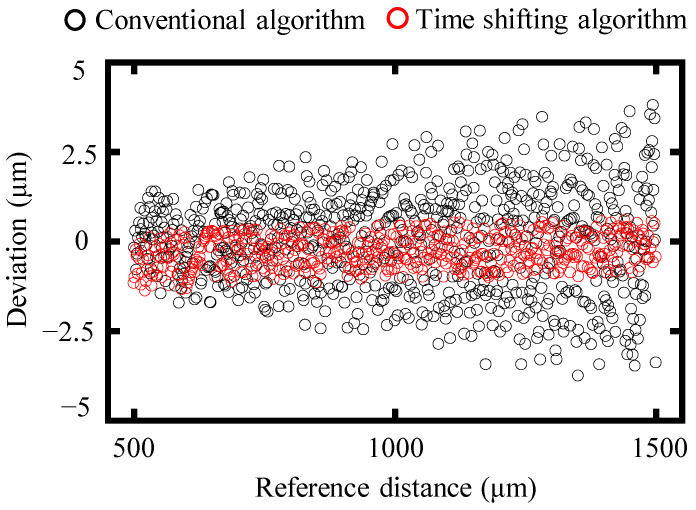
A comparison of the deviation of simulation results from the reference distance of the time-shifting and the conventional algorithm. The black and red points represent the simulated deviations of the conventional and time-shifting algorithms, respectively.

**Figure 6 sensors-24-02869-f006:**
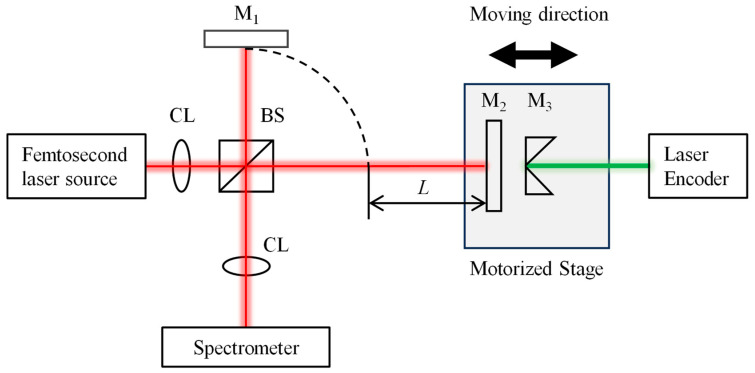
A schematic of the experimental setup for absolute distance measurements based on spectrally resolved interferometry. CL—collimating lens, BS—beam splitter, M_1_—reference mirror, M_2_—measurement mirror, M_3_—retroreflector, and *L*—length difference between the measurement beam and reference beam.

**Figure 7 sensors-24-02869-f007:**
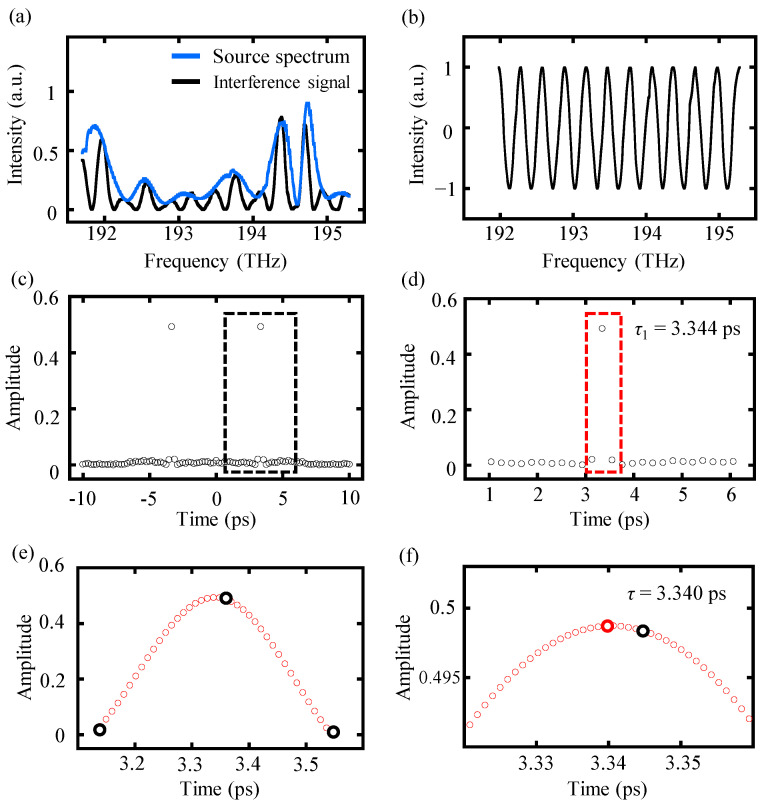
The data processing procedure of experimental data using the time-shifting algorithm. (**a**) The black line represents the raw experimental interference spectrum and the blue line represents the source spectrum, (**b**) the truncated interference signal after preprocessing, (**c**) inverse Fourier transform results of the truncated signal of (**b**) and the dot black frame highlights the selected time pulse *τ*_1_, (**d**) amplification of the results in the black frame of (**c**) and the red frame highlights the selected time pulse *τ*_1_ and its two adjacent time points, i.e., *τ*_1_ ± ∆*t*′, (**e**) inverse Fourier transform results of the time-shifting algorithm. Three black circles represent the inverse Fourier transform results of the original three time points in the red frame of (**d**) obtained by the conventional algorithm. The red circles represent the inverse Fourier transform results of a sequence of time points near the selected time pulse *τ*_1_, obtained by the time-shifting algorithm, (**f**) amplification of the central region of (**e**). The real-time delay *τ*, highlighted as the block red circle, is positioned by searching for the maximum absolute value of the inverse Fourier transform results among the time-shifting sequence.

**Figure 8 sensors-24-02869-f008:**
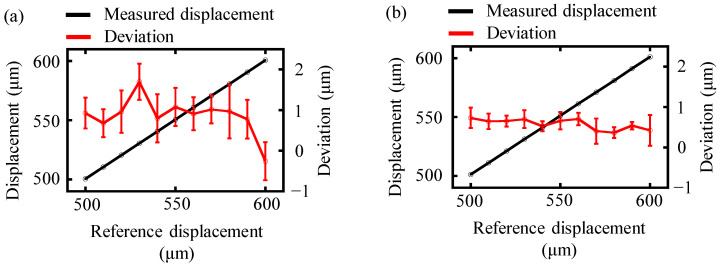
A comparison of the experimentally measured displacement of the conventional and time-shifting algorithm. The black and red lines represent the measured displacement and deviation from the reference distance, respectively. (**a**) The measurement displacement and the deviation from the reference distance of the conventional algorithm, and (**b**) the measurement displacement and the deviation from the reference distance of the proposed time-shifting algorithm.

**Table 1 sensors-24-02869-t001:** Summary of uncertainty sources for the refractive index.

Symbol	Standard Uncertainty	Sensitivity Coefficient	|*c*| · |*u*|
*u_t_*	0.015 K	9.3 × 10^−7^	1.395 × 10^−8^
*u_p_*	0.038 hPa	2.7 × 10^−9^	1.026 × 10^−10^
*u_h_*	0.204%	8.6 × 10^−9^	1.754 × 10^−9^
*u_n_*			1.406 × 10^−8^

**Table 2 sensors-24-02869-t002:** Summary of uncertainty sources for the distance *L*.

Source of Uncertainty	Symbol	Standard Uncertainty	Sensitivity Coefficient	|*c*| · |*u*| [m]
*n*	*u_n_*	1.406 × 10^−8^	−5.013 × 10^−4^ m	7.048 × 10^−12^
*τ* _1_	*u_τ_* _1_	3.176 × 10^−4^ ps	1.498 × 10^8^ m/s	4.756 × 10^−8^
*t_s_*	*u_ts_*	3.336 × 10^−4^ ps	1.498 × 10^8^ m/s	4.997 × 10^−8^
*cov*(*τ*_1,_*t_s_*)		−2.914 × 10^−30^ s^2^	2 · *c_τ_*_1_ · *c_ts_* · *cov*(*τ*_1_, *t_s_*)	−1.305 × 10^−13^
Combined uncertainty *u_L_*				3.550 × 10^−7^

## Data Availability

The data presented in this study are available on request from the corresponding authors.

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
