# Peer review of "An Improved Data Processing Algorithm for Spectrally Resolved Interferometry Using a Femtosecond Laser"

_sensors, 2024, doi:10.3390/s24092869_

Round 1

Reviewer 1 Report

Comments and Suggestions for Authors

This manuscript describes a new data processing algorithm for spectrally resolved Michelson interferometry. A new and optimized time shifting method  has been developed which demonstrates that a deep understanding of the mathematics of (inverse) Fourier transformation can result in a simple but significant reduction of measurement uncertainties. The manuscript is well written and the underlying mathematics in data processing is perfectly explained. Also the description of uncertainties by GUM is presented in an ideal manner and should be a good example for many other papers in this area. The manuscript should, thus, be published as it is.

Author Response

Thanks reviewer for highly evaluating our manuscript.

Reviewer 2 Report

Comments and Suggestions for Authors

Reviewer 3 Report

Comments and Suggestions for Authors

The introduction and literature review are strong. The work is well explained, and the figures are nicely presented. I do not have expertise in this area, but I found it interesting to read, and I'm sure it will be a useful contribution to the literature.

Author Response

(The authors gave the same response as above.)

Reviewer 4 Report

Comments and Suggestions for Authors

In this paper, a time shifting algorithm is proposed as a data processing method to improve the accuracy of spectrally resolved interferometry for distance measurement. Simulation and experimental results both demonstrate the advantage of the proposed method for improving the accuracy for distance sensing. A low average deviation of 0.58um respective to the reference distance is verified using the proposed algorithm and the expanded uncertainty of absolute distance measurement is obtained as 0.138um with a 95% confidence. In my point of view, the ideal of the proposed data processing method is new and the results shown in the manuscript are solid. The manuscript can be accepted after addressing the following concerns:

1.       Since the deviation of the distance measurement using spectrally resolved interferometry arises from the non-ideal truncation of the interference spectrum due to the limited spectral resolution of spectrometers, have the authors tried to interpolate the measured interference spectrum to improve the spectral resolutions so that a more ideal truncation of the spectrum can be made? I suggest the authors comment on the comparison of interpolation method and the proposed method.

2.       How efficient is the proposed method as I noticed that iFFT has to be performed for each slice of the time points with different time shifts?
